# *Alternaria* as an Inducer of Allergic Sensitization

**DOI:** 10.3390/jof7100838

**Published:** 2021-10-07

**Authors:** Guadalupe Hernandez-Ramirez, Domingo Barber, Jaime Tome-Amat, Maria Garrido-Arandia, Araceli Diaz-Perales

**Affiliations:** 1Centro de Biotecnología Y Genómica de Plantas (CBGP, UPM-INIA), Universidad Politécnica de Madrid (UPM), Instituto Nacional de Investigación y Tecnología Agraria y Alimentaria (INIA), 28223 Madrid, Spain; guadalupe.hernandez@upm.es (G.H.-R.); jaime.tome@upm.es (J.T.-A.); maria.garrido@upm.es (M.G.-A.); 2Departamento de Biotecnología-Biología Vegetal, Escuela Técnica Superior de Ingeniería Agronómica, Alimentaria y de Biosistemas, Universidad Politécnica de Madrid (UPM), 28040 Madrid, Spain; 3Departamento de Ciencias Médicas Básicas, Facultad de Medicina, Instituto de Medicina Molecular Aplicada (IMMA), Universidad San Pablo CEU, CEU Universities, 28925 Madrid, Spain; domingo.barberhernandez@ceu.es

**Keywords:** mold asthma, *Alternaria alternata*, *Alternaria* allergens, Alt a 1

## Abstract

*Alternaria alternata* is a saprophytic mold whose spores are disseminated in warm dry air, the typical weather of the Mediterranean climate region (from 30° to 45°), with a peak during the late summer and early autumn. *Alternaria* spores are known to be biological contaminants and a potent source of aeroallergens. One consequence of human exposure to *Alternaria* is an increased risk of developing asthma, with Alt a 1 as its main elicitor and a marker of primary sensitization. Although the action mechanism needs further investigation, a key role of the epithelium in cytokine production, TLR-activated alveolar macrophages and innate lymphoid cells in the adaptive response was demonstrated. Furthermore, sensitization to *A. alternata* seems to be a trigger for the development of co-sensitization to other allergen sources and may act as an exacerbator of symptoms and an elicitor of food allergies. The prevalence of *A. alternata* allergy is increasing and has led to expanding research on the role of this fungal species in the induction of IgE-mediated respiratory diseases. Indeed, recent research has allowed new perspectives to be considered in the assessment of exposure and diagnosis of fungi-induced allergies, although more studies are needed for the standardization of immunotherapy formulations.

## 1. Introduction

Fungal spores are an abundant component of the atmosphere, constituting the largest proportion of aerobiological particles in the environment [1]. However, of more than 100,000 known fungal species, only a few are responsible for causing airway diseases [2], and most of them belong to the genus *Alternaria. Alternaria* is the most potent sensitizing aeroallergen source [3], with an important impact on agriculture and health. In the case of agriculture, it is responsible for major crop losses as a plant pathogen [4] and, in terms of human health, it is strongly linked to respiratory disorders, such as asthma, rhinosinusitis [5], pneumonitis, skin infections, and bronchopulmonary mycosis [6,7,8]. Asthma is the most severe disease associated with *Alternaria* [9,10]. It is characterized by impaired lung function and frequent situations that can lead to death [11]. There is ample evidence demonstrating the unequivocal association among sensitization to *A. alternata*, the severity of asthma, and hospital admissions to intensive care units [10].

In a broad ISAC-based screening (Thermo Fisher, Uppsala, Sweden) in more than 1000 patients included in grass clinical trials in North America, 22% of the patients were sensitized to mold, and most of them recognized *Alternaria* [12], suggesting that the clinical relevance of this mold is underestimated.

The goal of this review is to detail the current information regarding *A. alternata*-induced sensitization and immune responses, with a special focus on the role of its major allergen, Alt a 1.

## 2. *Alternaria alternata* as a Plant Pathogen

As a plant pathogen, *A. alternata* causes black spots on many fruits and vegetables, including cereals, oilseeds, tomatoes, cucumbers, oranges, apples, etc., compromising almost 100 plant species [13]. As a saprophyte, it can infect senescing plants, but it also behaves as a latent fungus that can develop during the cold storage of fruits or remain quiescent for weeks until the fruit ripens.

The genus *Alternaria* is comprised of more than 350 species which may be saprophytic, endophytic, or even pathogenic in nature [14]. This wide range of action is partly due to the ability of Alternaria to survive in low nutrient media and temperatures ranging from −3 to 35 °C [15]. Its presence has been described in countries in the temperate zone of the planet, such as China, Europe, the United States, Australia and New Zealand [16,17], although there are few data from Africa [18].

*Alternaria* is usually described based on morphology and/or host specificity. Nowadays, molecular phylogenetic studies have provided a new classification system of twenty-seven sections based upon seven nuclear genes with Alt a 1 and plasmatic membrane calmodulin loci among those genetic marker [19].

## 3. *Alternaria alternata* in Human Health

The greatest risks in the case of fungal exposure are the development of respiratory diseases and hypersensitivity. Although hundreds of species that belong to the genus *Alternaria* are described, only two of them are linked to the development of allergies, i.e., *A. alternata* and *A. chartarum* (*Ulocladium chartarum*) (allergen database, http://www.allergen.org/; accessed on 20 July 2021).

The huge variation in the published data about the prevalence of *Alternaria alternata* among allergic patients means that it is very difficult to estimate the real prevalence of *Alternaria* sensitization. This uncertainty may be mainly because of the lack of standardization of extracts for diagnosis [20], until very recently.

In any case, in the literature, several studies can be found where *Alternata* sensitization is studied among the allergic population. Results from the European Community Respiratory Health Survey I (1990–1992) conducted in 37 centers in 17 countries, involving countries with different geographical parallels such as Australia, the United States of America (USA), the United Kingdom (UK) and Iceland, showed a mean prevalence among populations with respiratory allergic diseases of 4.4% sensitization to *A. alternata*, varying from 0.2% to 14.4% in the different centers [21,22].

Similarly, a multicenter and open-label study conducted by the European Asthma and Allergy Network (GA2LEN) in 14 European countries have estimated that the prevalence of *Alternaria* sensitization was, on average, around 6.1%, varying from 2% in Finland to 20% in Greece, in 2009 [23]. In Asia, a cross-sectional study in Iran with about 1000 allergic patients is noteworthy. In this study, sensitization was found to be 5.3%, based on skin prick test reactivity [24]. Finally, in China (Guangdong), sensitization to *A. alternata* among asthmatic children was 14.9%, while among asthma patients it was 44.9% [16].

These results are in line with the results obtained in the German population, analyzing fungal sensitization in patients with respiratory diseases over 20 years (1998–2017). It was shown that, in the second decade of life, the percentage of fungal sensitization doubled as compared with in the first decade of life [25].

Finally, it is worth mentioning that people working in farms and sawmills are at high risk of *A. alternata*-induced allergic reactions and infections due to the high level of its allergen in these environments. In addition, among people with respiratory allergy, younger individuals are at higher risk of developing sensitization to *A. alternata* than older people [5].

## 4. *Alternaria alternata* Allergens

Taking into consideration the association between *Alternaria* and asthma severity [5], a complete definition and immune characterization of the allergen repertoire would contribute by increasing our understanding of this mold as a powerful respiratory allergic disease inducer. To date, a total of seventeen proteins are characterized as allergens in *A. alternata* (http://www.allergen.org/, accessed on 1 October 2021 and http://www.allergome.org/, accessed on 1 October 2021). This list includes proteins restricted to a small number of taxonomically related fungal species and ubiquitous proteins that were conserved throughout the evolutionary process (Table 1). Most of them have homologues in the other three relevant mold genera in allergy: *Cladosporium, Penicillium* and *Aspergillus*, with the exceptions of Alt a 1 and Alt a 13, the most relevant allergens.

**Table 1 jof-07-00838-t001:** *Alternaria alternata* allergens.

Allergen	Clinical Relevance	Biological Function	MW (kDa)	Protein (UNIPROT)	Reference
Alt a 1	Major (95%)	Unknown	16.4 and 15.3	P79085	De Vouge et al., 1998 [26]
Alt a 2	Inconclusive	Unknown	22.3–25	O94095	Kustrzeba-Wójcicka et al., 2014 [27]
Alt a 3	Minor (5%)	Heat shock protein 70	70	P78983	De Vouge et al., 1998 [26]
Alt a 4	Minor (42%)	Disulfide isomerase	57	Q00002	Achatz et al., 1995 [28]
Alt a 5	Minor (8–14%)	Ribosomal protein P2	11	P42037	Achatz et al., 1995 [28]
Alt a 6	Minor (22%)	Enolase	45	Q9HDT3	Simon-Nobbe et al., 2000 [29]
Alt a 7	Minor (7%)	Flavodoxin, YCP4 protein	22	P42058	Achatz et al., 1995 [28]
Alt a 8	Minor (41%)	Mannitol dehydrogenase	29	P0C0Y4	Schneider et al., 2006 [30]
Alt a 9	Minor (5%)	Unknown	43	-	Kustrzeba-Wójcicka et al., 2014 [27]
Alt a 10	Minor (2%)	Aldehydede hydrogenase	53	P42041	Achatz et al., 1995 [28]
Alt a 12	Minor	Acid ribosomal protein P1	11	P49148	Achatz et al., 1995 [28]
Alt a 13	Minor (82%)	Glutathione-S-transferase	26	Q6R4B4	Shankar et al., 2006 [31]
Alt a 14	Minor (11.5%)	Manganese superoxide dismutase	24	P86254	Kustrzeba-Wójcicka et al., 2014 [27]
Alt a 15	Minor (10.2%)	Vacuolar serine protease	58	A0A0F6N3V8	Kustrzeba-Wójcicka et al., 2014 [27]
Alt a NFT2	Minor (0.8%)	Nuclear transport factor 2	13.7	Q8NKB7	Weichel et al., 2003 [32]
TCTP	Minor (4%)	Translationally controlled tumor protein	20–30	D0MQ50	Rid R., et al., 2009 [33]
Alt a 1 70 kDa	Minor (87%)	Unknown	70	-	Olson et al., 1990 [34]

While more than 90% of patients sensitized to *Alternaria* had positive skin test to Alt a 1 [35], it was considered the only well-defined major allergen of this source. However, Alt a 13 is suggested to be another major allergen because of eliciting skin reactions in 14 of 17 patients [31]. Alt a 13 is a 26 kDa glutathione-S-transferase whose function is the detoxification of endogenous and xenobiotic compounds by conjugation of these compounds to reduce glutathione. It is recognized as a cross-reactive allergen in fungal extracts [36]. However, there are insufficient studies, and the studies performed to support its classification as a major allergen were carried out with a small number of patients.

### Alt a 1 and Its Flavonol Ligand

Alt a 1 (AAM90320.1, NCBI Protein Database) is a small protein mainly detected in mold spores, which is released at the beginning of germination [37]. Although it is included in its own protein family (AA1) with unknown activity, studies on its homologous Alt b 1 in *A. brassicicola* and its capacity of inhibiting plant proteins related to pathogenesis (PR5) suggest a biological role in pathogenicity [38,39,40,41]. Other studies support that Alt a 1-like proteins play a key role in the development of plant–fungal interactions [42,43].

In addition, Alt a 1 is also well characterized as a major allergen and marker of primary sensitization [43]. It is strongly associated with chronic asthma and is declared as a risk factor for the development and exacerbation of asthma. Its high-resolution X-ray crystal structure reveals a unique β-barrel fold formed by eleven β chains that are reported to have no equivalent in the Protein Data Bank [44,45]. The structure is stabilized as a heat-stable, 30 kDa homodimer that dissociates into 15 KDa subunits under reducing conditions or acidic pH. The homodimer shows a “butterfly” shape (Figure 1a) stabilized by an intermolecular disulfide bridge between Cys30 from both monomers and by hydrophobic and polar interactions [44].

In Alt a 1, two linear IgE-binding epitopes (K41-P50 and Y54-K63) are identified [46]; both are localized, exposed on the dimer [47] (Figure 1b), and allow the IgE interaction. In contrast, the tetrameric oligomerization of Alt a 1 (Figure 1c), stabilized by its native ligand, a methyl derivative of quercetin [37,44], hides these residues (Figure 1d), blocking the interaction with the IgE antibodies.

## 5. Immunological Mechanism of *Alternaria* Action

Several studies have demonstrated that *Alternaria* extract is able to provoke allergic responses in mice; the extract is composed of a complex mixture of proteins, mycotoxins, cell wall fragments, chitin, mannans, and B-1,2-glucans [48]. However, there are no data concerning the contribution of each specific component of this mixture in proinflammatory responses. Within this context, the current information about the innate and adaptive immune responses associated with *Alternaria* action are summarized further with special attention to the role of Alt a 1 in allergy triggering. An overview of the events that take place is described in this section and is shown in Figure 2.

### 5.1. First Line of Defense: The Early Response

The human respiratory epithelium is a mucosal surface composed of ciliated cells, mucous-producing cells, and undifferentiated basal cells. It is the first site of contact for all inspired substances, acting as both a physical and an immunological barrier to pathogens and external particles to aid the maintenance of normal airway function [49]. Therefore, when mold spores are inhaled and reach the upper airways, the external epithelial barrier responds by producing antimicrobial molecules, proinflammatory cytokines, and chemokines for immune cell recruitment [50]. Many studies on the effects of fungal extracts in the lungs have focused on how they are recognized by pattern recognition receptors (PPRs) and/or toll-like receptors (TLRs) [51,52,53].

*Alternaria,* by itself, can induce the activation of the epithelium, inducing a quick release of alarmins such as thymic stromal lymphopoietin (TSLP), IL-25, and IL-33, and even the production of other inflammatory cytokines such as IL-6 and IL-8 [54,55,56,57,58,59]. Likewise, other cellular responses are associated with *Alternaria*, such as ATP release, calcium signaling, calprotectin production, or eosinophil activation [60,61]. Interestingly, some of these studies were conducted using heat inactivated *Alternaria* extracts, which suggests that protease activities are not necessary to induce asthma pathogenesis by this mold.

Alt a 1 is released in large quantities by spores at pH values around 5.0–6.5, carrying its flavonol and catechol ligands [37]. On the one hand, Alt a 1 interacts with the SLC22A17 receptor in bronchial epithelial cells and mediates the production of IL-33 and IL-25 [62]. On the other hand, this interaction does not result in an increase in barrier permeability, and thus does not explain how Alt a 1 reaches the basolateral side and comes in contact with the immune system cells.

Indeed, Hayes et al. [53] reported that Alt a 1 induced the production of IL-8, MCP-1, and Groa/b/g in airway epithelial cells in a TLR2/4-dependent manner, suggesting that Alt a 1 could be recognized by different subsets of immune cells such as alveolar macrophages. These immune cells, together with the epithelial barrier, are the first line of defense in lungs, by constantly sensing the local environment [63,64]. Therefore, given the well-known function of alveolar macrophages, they could contribute to Alt a 1 incorporation and processing, triggering the production of proinflammatory cytokines. This inflammatory environment, together with the presence of alarmins from epithelial cells, may further impair the epithelial barrier. Moreover, these cells could be implicated in the presentation of Alt a 1 to adaptative immune cells [65].

### 5.2. Type 2 Immune Response: The Adaptative Response

Asthma and respiratory allergy are two diseases that are still not fully understood, with several unrelated explanations for the mechanisms of how a patient acquires these phenotypes. However, numerous studies have reported a strong correlation between *Alternaria* inhalation and the development of asthma or allergic rhinitis [66,67,68,69].

*A. alternata* promotes the production of IL-4, IL-10, and IL-6 in antigen-presenting cells (APCs), together with an increase in the expression of MHC class-II, CD40, CD80, and OX40L, which are costimulatory molecules implicated in type 2 development, in in vitro approximations [70]. Moreover, *Alternaria* can inhibit the endosome receptor TLR3 and the production of IFN-beta, IP-10, and I-TAC to inhibit defense mechanisms for viral and fungal infections [71].

In the literature, there are many murine models, in which the application of extract or only spores show airway hyperreactivity, mucus hypersecretion, humoral response, lung inflammation, and eosinophilic infiltration, among other effects related to airway diseases. Furthermore, in these models, the production of type 2 cytokines IL-4, -5, and -13, as well as high levels of *Alternaria* specific IgE is also observed [54,72].

In the last decade, many studies have focused on the critical role of innate lymphoid cells type 2 (ILC2s) in *Alternaria*-mediated immunopathology in the airways [59,69,73,74]. Tissue resident ILC2s are described as highly responsive, early effectors in type 2 responses, functioning as a prominent source of typical type 2 cytokines IL-5, IL-9, and IL-13, long before an adaptative response is initiated [75]. By studying the role of this cell population in mice exposed to *Alternaria*, a rapid type 2 cytokine response driven by an activated lung ILC2 population has been established. Therefore, the restoration of lung pathology in mice deprived of ILC2s and exposed to *Alternaria* has finally confirmed the contribution of ILC2s to allergic airway inflammation in animal models [54,76,77].

In summary, *Alternaria* can induce a type 2 response, resulting in a remarkable increase in the number of eosinophils, accompanied by an increase in APCs (such as ILCs) and effector cells such as mast cells [78,79].

## 6. The Trojan Horse: *Alternaria* as an Inductor of Other Allergies

Airborne spore counts are often 1000-fold greater than pollen counts, and exposure is often longer in duration. Indeed, there is a clear correlation between high atmospheric concentrations of certain mold spores (for instance during thunderstorms) and an increase in hospital admissions due to asthma exacerbations [80]. In addition, many studies have noted that asthma severity is often more closely associated with *Alternaria* sensitization than with pollen [81]. Within the clinical relevance of *Alternaria*, its potential as an indoor mold should not be underestimated, since it can grow in dampness and even on fruits and vegetables without visible signs of infection [41]. Thus, it is reasonable to consider that molds such as *Alternaria* may be initiators of allergic responses, either food and/or respiratory diseases. The next sections focus on two relevant examples regarding co-sensitization events with *A. alternata*, which could aid in understanding and predicting sensitization and improving allergy diagnostic methods.

### 6.1. Alternaria and Grass Pollen Cosensitization

Continuously changing environments lead to differences in exposure and sensitization to allergens, which in turn vary depending on geographic region [80]. For example, the monitoring of *Alternaria* conidia in 12 stations in Spain showed three different patterns of spore dynamics and concentration levels correlated to daily temperatures and precipitation [82]. Likewise, correlations between *Alternaria* spore counts and symptoms of rhinitis and/or asthma have been described for *Alternaria*-sensitized patients [83]. Thus, the development of logistic regression models for predicting the daily concentrations of airborne spores could also aid to predict symptoms or improve the diagnosis of fungi-induced airway diseases [82]. Hernandez-Ramirez et al. [84] described a co-exposure event for grass pollen-allergic patients based on a correlation between high levels of *Alternaria* spores and asthmatic exacerbations suffered by the patient population when pollen was no longer present. After the identification of Phl p 1-like proteins in grass straw and their ability to bind over the surface of *Alternaria* spores, the authors suggested that mold spores may act as allergen carriers favoring the development of co-sensitization between *Alternaria* and other aeroallergens-like group 1 allergens (Figure 3a). It has long been described that the levels of respirable particles, including pollen grains and fungal spores, increased following rainfall events, as well as the release of their allergenic components as a result of the force of storms, moisture conditions, and/or disturbances such as wind and lawn mowing [85]. Interestingly, when the sensitization profile was studied in patients with thunderstorm asthma, the clear co-sensitization to both aeroallergen sources suggested that more than one hit was required [81].

Furthermore, the role of *Alternaria* as a clinically relevant model for environmental exposure was also studied in asthma developed by ragweed pollen allergy. Consistent with previous reports that showed the adjuvant effects of *Alternaria* to induce a potent type 2 response in OVA-sensitized mice [86], Kobayashi et al. [87] obtained the same result for mice sensitized to short ragweed pollen, whose humoral responses were minimal when they were exposed to the extract alone. In this study, the authors also introduced the key role of DCs exposed to *Alternaria* to enhance type 2 polarization of CD4^+^ T cells. Likewise, a single challenge with *Alternaria* extract resulted in increased eosinophils, peribronchial inflammation, and mucus production, as well as type 2 cells and ILC2s recruitment to the airway in ryegrass challenged mice, suggesting that the combination of mold spores and increased pollen allergen exposure during thunderstorms may be responsible for severe asthma [57].

### 6.2. Alternaria and Kiwifruit Cosensitization

In addition to *Alternaria* effects as airborne mold, *Alternaria* spp. are also well known to infect a wide variety of fruits and vegetables [4]. *Alternaria* contamination is also documented to trigger food allergy, as reported by Gómez-Casado et al. [41] In controlled infection assays of kiwifruit with *Alternaria* spores, the authors demonstrated that Alt a 1 was released from spores to kiwifruit pulp, partially inhibiting the activity of a defense plant protein, the PR5 thaumatin-like protein, when hyphal growth was not yet developed. This study found the origin of a co-sensitization phenomenon between Alt a 1 and Act d 2, a major allergen and PR5 of kiwifruit, caused by the consumption of kiwifruits infected with *Alternaria* but apparently in good conditions (Figure 3b). According to the early detection of Alt a 1 in infected kiwifruits, similar results were obtained in citrus fruits, highlighting the value of Alt a 1 as a reliable and specific marker of fungal contamination [88].

At the same time, the *Alternaria*-spinach syndrome was also reported in mold-sensitized patients with food allergic reaction after mushroom (*Agaricus bisporus)* and spinach ingestions [89,90].

Accordingly, food contamination with fungal spores may be the cause of some adverse reactions that patients suffer from, who apparently were not sensitized in the past, and obtained negative results in previous skin prick tests for the diagnosis of a food allergy.

## 7. Immunotherapy for *Alternaria* Sensitization

There is limited evidence on the benefit of allergen immunotherapy (AIT) in *Alternaria* allergy, the only mold allergy with high prevalence [91]. The European Academy of Allergy and Clinical Immunology (EAACI) does not recommend AIT using mold extracts in children. This opinion was based on a lack of efficacy and safety [92]. On the other hand, The American Academy of Allergy, Asthma and Immunology (AAAAI) has concluded that such an AIT might be effective. An international consensus on immunotherapy indicated the possibility of AIT for mold allergy, but only with standardized extracts [93].

This type of allergy is relatively frequent in children under 10 years of age as a mono-sensitizing allergen [94,95]. In adults, *Alternaria* sensitization is mostly associated with another allergy, thus making AIT questionable. With current regulations, performing clinical trials in pediatric patients requires previous clinical experimentation in adults. Moreover, in many cases, the asthma associated with *Alternaria* might be a contraindication, hence the design of appropriate clinical trials is difficult and expensive, a fact that has limited the building of clinical evidence. Traditionally, the active ingredients of any AIT product have consisted of protein mixtures [96], causing, for instance, great variability in the concentration of allergens in the existing products for in vivo allergy diagnosis and AIT [20]. Assuring the correct content of Alt a 1 in AIT preparation is critical to initiate any AIT intervention in patients with *Alternaria* allergy, as was shown in an interesting therapeutic approach with positive initial results based on the use of purified Alt a 1 in AIT preparation [35].

Many studies have highlighted the effectiveness of AIT for *Alternaria* in patients with allergic rhinitis and/or bronchial asthma [12,97,98,99,100,101] This applies to children and adults. This is the only mold allergen for which medical documentation has confirmed a significant improvement after treatment in controlled studies. The effectiveness of such AIT is dependent upon the quality of the vaccine. The development of methods for the extraction and assessment of the main allergen, Alt a 1, has occurred because standardization made it possible to extract *Alternaria alternata*. Studies that used a vaccine containing the standardized *Alternaria alternata* extract have confirmed the effectiveness of such an AIT [102]. This treatment led to reduced clinical allergic rhinoconjunctivitis symptoms, bronchial asthma, reduced drug used, decreased serum specific IgE levels, and increased serum specific IgG4 levels to *Alternaria*. It was also important to specify the cumulative Alt a 1 antigen doses that patients received during AIT in these trials, since this was one of the criteria for assessing the effectiveness of AIT. These authors simultaneously observed very good tolerance of such vaccines. Adverse events were observed in less than 1% of all injections. There were mainly local reactions and only a few systemic reactions, which included dyspnea, cough, and rhinitis [103,104,105]. Earlier, an older vaccine (without proper allergen extract standardization) induced many adverse systemic reactions. This greatly limited the possibilities of such treatment [106,107,108].

Despite the effectiveness of AIT treatment with *Alternaria*, there is a lack of long-term follow up of patients after such treatment. There is no feedback on how long the effect persists after desensitization to *Alternaria*. Long-term observations of the effectiveness of AIT are difficult and expensive to implement.

## 8. Conclusions

Fungal allergies are increasing in recent years, and thus it is essential to understand and describe the molecular events associated with them. In addition, symptomatology is often associated with the development of severe hypersensitivity reactions in the airways that compromise the health and quality of life of susceptible individuals.

One of the main fungi responsible for this pathology is *A. alternata*, and although it is traditionally described as one of the main causes of childhood asthma, its prevalence is also increasing in adults. Because of this increase, a comprehensive understanding of the mechanisms and diagnosis of allergic symptoms caused by the mold are necessary to acknowledge why and how it can induce lung inflammation prior to the development of asthma.

However, despite efforts to describe the potential of *Alternaria* as an inducer of respiratory allergic inflammation, questions about the individual effects of its allergenic components, especially Alt a 1 as the major allergen, remain unresolved. It is only by means of a complete characterization of immunological mechanisms triggered by this mold that we can improve the diagnosis and, above all, the design of new treatment formulations.

## Figures and Tables

**Figure 1 jof-07-00838-f001:**
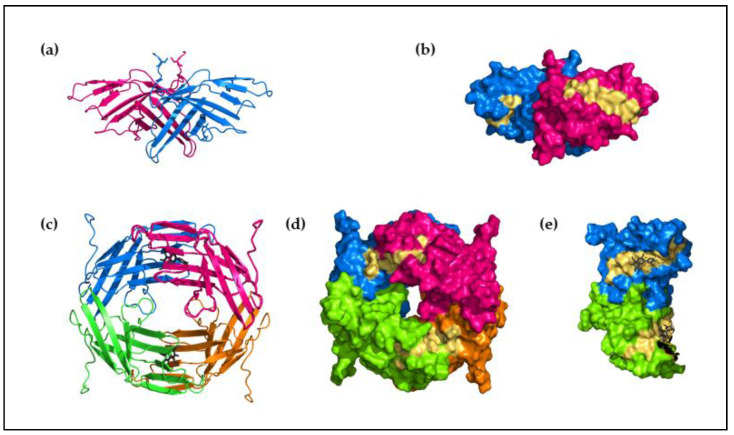
Protein structure of Alt a 1: (**a**) Schematic diagram of the dimeric structure of Alt a 1 in its “butterfly” shape; (**b**) molecular surface of the dimeric structure of Alt a 1. Chains are shown in different colors and the position of the epitopes are shown in yellow, the epitopes are oriented to the outside of the structure leading the interaction with IgE antibodies; (**c**) schematic diagram of the tetrameric structure of Alt a 1; (**d**) molecular surface of the tetrameric structure of Alt a 1, chains are shown in different colors and the epitopes in yellow; (**e**) detail of the tetrameric molecular surface, showing the IgE binding epitopes (yellow) buried in the structure. PDB code, 3v0r.

**Figure 2 jof-07-00838-f002:**
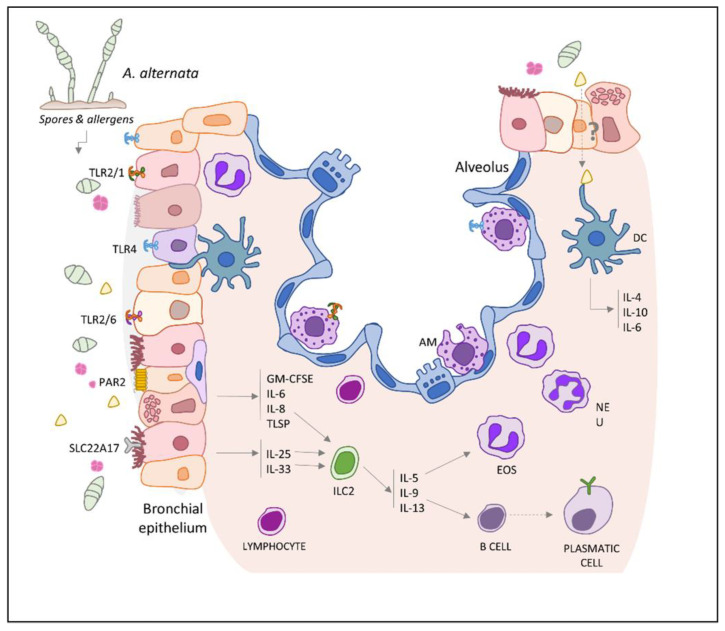
Immune responses induced in airway epithelium by *Alternaria alternata*. Spores and mycelium fragments containing allergenic compounds (such as Alt a 1) can reach upper airways, where they can interact with TLRs (TLR2, TLR4), PAR 2, or SLC22A17 receptors present in the alveolar macrophages and epithelial cells. The activation of these types of cells induces proinflammatory responses by means of the production of alarmins (IL-33, IL-25, TSLP) and other proinflammatory cytokines. Alarmins promote the recruitment and activation of innate lymphoid cells type 2 (ILC2), which have an important role in the development of *Alternaria*-induced type 2 responses (e.g., eosinophil infiltration, mucus hypersecretion, airway hyperreactivity, IgE production, etc.). EOS, eosinophil; NEU, neutrophil; DC, dendritic cell; AM, alveolar macrophage.

**Figure 3 jof-07-00838-f003:**
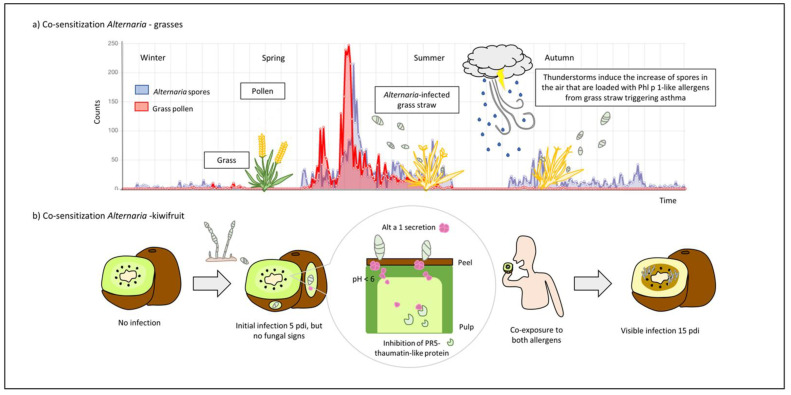
Co-sensitization phenomena between *Alternaria* and other allergens: (**a**) asthma exacerbation by sensitization to *Alternaria* grass pollen. In Central Spain, grass pollen patients suffer from asthma during late summer and early fall, when pollen is no longer present. In contrast, in this season, *Alternaria* spore peaks were registered (data from Ciudad Real, 2020, SEAIC, https://www.polenes.com/es/home, accessed on 1 October 2021). Considering that thunderstorms and grass postharvest are common at this time, the authors described the role of spores as Group-1 grass allergen carriers and suggested that the inhalation of loaded spores may be the cause of the asthma exacerbations in grass pollen patients; (**b**) *Alternaria*, kiwifruit co-sensitization. *Alternaria* infects kiwifruits secreting Alt a 1 that can be detected in pulp at 5 days post infection (dpi) while fungal hypha is not visible. The partial inhibition of a PR5 thaumatin-like protein (Act d 2) by Alt a 1 was reported, evidencing the role of the fungal protein in plant pathogenesis, and providing an explanation for the co-relation between Alt a 1 and Act d 2 sensitizations as a result of consumption of *Alternaria*-infected kiwifruits but apparently in good conditions.

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
