# Peer review of "Alternaria as an Inducer of Allergic Sensitization"

_jof, 2021, doi:10.3390/jof7100838_

Round 1

Reviewer 1 Report

  1. The authors summarize the current state of knowledge on the allergens of Alternaria alternata, its immunological mechanism, and two examples as inductor of other allergies. The MS is informative.
  2. The topic is appropriate for publication in Journal of Fungi, because respiratory tract is regarded as a natural spore trap where airborne pathogenic fungi are deposited, of which research is important, emerging scientific task.
  3. However, authors intersperse with plant pathology of Alternaria and A. alternata among the review without well connection to the goal. For example, under the goal (Line 49. 'The goal of this review is to detail the current information about A. alternata-induced sensitization and immune responses, with special focus on the role of its major allergen, Alt a 1.'), please explain whether and how the phytotoxins (Line 65) and mycotoxins (Line 77) associate with allergic sensitization? Readers will expect to know about it.
  4. Only few comments and controversies were addressed. If there are more critique and evaluation which readers would benefit from reading the article.
  5. The English language and grammar needs extensive revision before the manuscript can be accepted for publication. For example:

Line 42. 'characterised'

Line 117. 'the IgE-binding epitopes of'   of what?

Line 119. 'haven been'

Author Response

Response to Reviewer 1 Comments

The authors summarize the current state of knowledge on the allergens of Alternaria alternata, its immunological mechanism, and two examples as inductor of other allergies. The MS is informative.

The topic is appropriate for publication in Journal of Fungi, because respiratory tract is regarded as a natural spore trap where airborne pathogenic fungi are deposited, of which research is important, emerging scientific task.

Point 1:  However, authors intersperse with plant pathology of Alternaria and A. alternata among the review without well connection to the goal. For example, under the goal (Line 49. 'The goal of this review is to detail the current information about A. alternata-induced sensitization and immune responses, with special focus on the role of its major allergen, Alt a 1.'), please explain whether and how the phytotoxins (Line 65) and mycotoxins (Line 77) associate with allergic sensitization? Readers will expect to know about it.

Response 1: The referenced paragraph (L765-L74) has been removed from the revised text , because as commented by the reviewer, as it is not related to the main topic of the manuscript. To the best of our knowledge, there is no published study linking the phytotoxin production to the mold ability to induce allergic responses.

Point 2:Only few comments and controversies were addressed. If there are more critique and evaluation which readers would benefit from reading the article.

Response 2: We agree with the reviewer. The new version has been deeply revised, improving several points. However, However, it is possible that some issues that are considered to be of interest have not been dealt with in sufficient depth. We would therefore be grateful if you could point them out to us in order to review them and include them in the revision.

Point 3:The English language and grammar needs extensive revision before the manuscript can be accepted for publication.

Response 2: The manuscript has been revised by the English editing service provided by the publisher.

Reviewer 2 Report

Generally, represented article is not bad and does not requires much improvement. However, some inappropriate expressions and some obsolete facts are given there.

In addition, corrections of English grammar and correct representation of Botanical names of plants is needed.

More information is in attached files.
Please, note that at corrections that pertain to the beginning of the text are given here.
Yet, starting from line 229, you will find this corrections in the pdf-file of your article.
Some of them, which I considered the most important, I also included into the file Word - just to be sure that information will not be lost. So, don't be surprised, if you find something repetitive, it is just for safety concerns. :)
I wish you to improve your article soon and have a good luck.

Thus:

  1. Check Italic for Latin names of genera and species representation, names of Alt a allergen starting from the Abstract and do it throughout the text, sometimes genera are not given in Italic, or their name doesn’t start from the capital letter (lines 15, 58, 99 and so on).
  2. English is not bad, but some expressions, e.g. lines 73-74, 99-101, 104-107 should be revised and corrected
  3. Modern scientific investigations, done in different parts of the world (Europe mostly) consider molecular data of sensitisation to different allergen sources. Some data are given in this review, but I would recommend to search and add more information, as it may change the interpretation of results.

Abstract and Introduction are generally ok, requiring just grammar and spelling checks

Chaper 2

Line 58, as you represent quantity by numbers, so 7 should be given as a number too (reading will be more consistent).

  1. Alternaria alternata in human health

Sphingolipid metabolism impairment is mentioned in the last paragraph of the previous chapter (line 74) and in the line 79, with no added value of information in the line 79, so, information just looks repetitive, 1 mention should be removed.

Lines 89-90 – probable, second decade of life is mentioned here, and I would recommend to change expression as follows for better understanding: «which shows that in the second decade of life the percentage of fungal sensitisation doubles compared to the first one».

Lines 99-101 is better to express as follows: “Given the aforementioned link between Alternaria and asthma severity [2], a full definition and immune characterization of the A. alternata allergen repertoire would contribute to improvement of our understanding of this mold as a powerful respiratory allergic disease inducer”.

Lines 104-106: “This list includes proteins restricted to a small number of taxonomically related fungal species and ubiquitous proteins that are conserved throughout the evolutionary process”.

Lines 112-113: It states: “While Alt a 1 is the only well-defined major allergen by being recognized by more than 90 % of patients” – recognized how? Allergen can be recognized as an allergen by the immune system of patient. Sentence needs revision.

Lines 117-118: It is given: “and the IgE-binding epitopes of have also been determined” – not clear epitopes of what? have also been determined, needs revision

Line 134 says: “Unlike the above mentioned”, but it looks like there should be written, “Like the above mentioned”, check, please.

Line 149 – word “stablish” is an obsolete variant of “establish”, better to use a newer variant.

Figure 1 – legend, sentence: “(e) Fragment of the molecular surface of the tetrameric 161 structure in which is shown (it is shown?) as the epitopes (epitope?), in yellow, are located buried in the structure blocking the interaction with the 162 IgE antibodies”, grammar needs revision, in general it is difficult to understand this expression

Chaper 5, better to change its title as follows: “Immunological mechanism of Alternaria action”

Lines 169 – 171: grammar should be checked. It is better to say: “Within this context, it the current information about the innate and adaptive immune responses associated with Alternaria action will be summarized further with special attention to the role of Alt a 1 in allergy triggering.

Line 177, instead of “foreign” it is better to use “alien or external”,

Lines 182-183 – is given: “Similar to the mechanisms before described for aeroallergens (i.e. house dust mite) [55], Alternaria can induce the activation of epithelium…”, grammar needs revision here, no suggestion as a couple of variants are possible there, anyway, sentence is grammatically incorrect.

Figure 2, legend: “Spores and 190 mycelium fragments containing allergenic compounds (such as Alt a 1 or proteases) can reach 191 upper airways and induce pro-inflammatory responses by means of activation (triggering? and so on) TLRs (TLR2, TLR4), PAR 2 or 192 SLC22A17 receptors present in epithelial cells

Lines 200-202, given: “heat inactivated Alternaria extracts showing that are not necessary protease activities to induce asthma pathogenesis”, better: “heat inactivated Alternaria extracts showing that protease activities are not necessary to induce asthma pathogenesis”. In case, it was mentioned something else, grammar needs more revision.

Line 203: “In the past years, more effort has been made to understand the role of different…” Probably, biological role?

Lines 206-207, it is given: “to find an explanation of the high prevalence of Alt a 1.” Prevalence where? Not clear.

Line 210 says: “In contrast, this interaction does not result in an increase in…” In the given context it is better to say: “On the other hand, this interaction does not result in an increase in”.

Line 229-300 – it is better to use “applied” instead of “exposed”.

See more in the attached file of the article text.

Grammar of entire subchapter, within lines 254-265, should be revised.

It is recommended to add more information about the evidences that Alternaria is a primary sensitizator. You mention it in the text, but evidences are not clear.

Lines 286-287: Grammar needs revision, it is not clear, which both sources are mentioned,

also, for epidemiology of thunderstorm asthma? Or some other word can be applied here, but "epidemics" seems irrelevant.

Lines 290-300 - Grammar needs revision here.

Fig. 3 should be given after line 324

Lines 341-358 - What is really interesting and even funny that the only world producer of AIT treatment for Alternaria is Inmunotek company, based in Spain.

Its AIT medication, including Oraltek drug developed to treat Alternaria alternata sensitivity is successfully sold in many European countries. As far as I know, they completed their clinical and other trials and are allowed to produce their medicine.

So, with no relation to the name of company, I am sure, you should change the information given in this paragraph. Up to now this information about non-availability of AIT for Alternaria sensitization is obsolete.

Lines 351-354 - Looks like this sentence fits better to the Introduction.

Author Response

Response to Reviewer 2 Comments (Please see the attachment)

Point 1:  Generally, represented article is not bad and does not requires much improvement. However, some inappropriate expressions and some obsolete facts are given there.

In addition, corrections of English grammar and correct representation of Botanical names of plants is needed.

More information is in attached files.

Please, note that at corrections that pertain to the beginning of the text are given here.
Yet, starting from line 229, you will find this corrections in the pdf-file of your article.
Some of them, which I considered the most important, I also included into the file Word - just to be sure that information will not be lost. So, don't be surprised, if you find something repetitive, it is just for safety concerns. :)
I wish you to improve your article soon and have a good luck.

Response 1: We are grateful to reviewer 2 for the help and suggestions to correct the entire text to improve its comprehensibility and enrich it.

Point 2: Check Italic for Latin names of genera and species representation, names of Alt a allergen starting from the Abstract and do it throughout the text, sometimes genera are not given in Italic, or their name doesn’t start from the capital letter (lines 15, 58, 99 and so on).

Response 2: All text has been reviewed changing for italic all Latin names, and starting by upper letter allergens.

Point 3: English is not bad, but some expressions, e.g. lines 73-74, 99-101, 104-107 should be revised and corrected

Response 3: The manuscript has been revised by the English editing service provided by the publisher.

Point 4: Modern scientific investigations, done in different parts of the world (Europe mostly) consider molecular data of sensitisation to different allergen sources. Some data are given in this review, but I would recommend to search and add more information, as it may change the interpretation of results.

Response 4: The section 'Alternaria alternata in Human Health' has been completed with new information.

Point 5: Abstract and Introduction are generally ok, requiring just grammar and spelling checks

Response 5: The manuscript has been revised by the English editing service provided by the publisher.

Chapter 2

Point 6: Line 58, as you represent quantity by numbers, so 7 should be given as a number too (reading will be more consistent).

Response 6: Thank you, Changes have been done.

Chapter 3: Alternaria alternata in human health

Point 7: Sphingolipid metabolism impairment is mentioned in the last paragraph of the previous chapter (line 74) and in the line 79, with no added value of information in the line 79, so, information just looks repetitive, 1 mention should be removed.

Response 7: All information about sphingolipid has been removed, due to its no-relationship with the principal aim.

Point 8: Lines 89-90 – probable, second decade of life is mentioned here, and I would recommend to change expression as follows for better understanding: «which shows that in the second decade of life the percentage of fungal sensitisation doubles compared to the first one».

Response 8: Thank you, change has been done as suggested.

Point 9: Lines 99-101 is better to express as follows: “Given the aforementioned link between Alternaria and asthma severity [2], a full definition and immune characterization of the A. alternata allergen repertoire would contribute to improvement of our understanding of this mold as a powerful respiratory allergic disease inducer”.

Response 9: Thank you, change has been performed.

Point 10: Lines 104-106: “This list includes proteins restricted to a small number of taxonomically related fungal species and ubiquitous proteins that are conserved throughout the evolutionary process”.

Response 10: Thank you, change have been done as suggested.

Point 11: Lines 112-113: It states: “While Alt a 1 is the only well-defined major allergen by being recognized by more than 90 % of patients” – recognized how? Allergen can be recognized as an allergen by the immune system of patient. Sentence needs revision.

Response 11: The sentence has been changed. Currently, it can be read:’ While Alt a 1 is the only well-defined major allergen that has been recognized by skin tests in more than 90% of patients sensitized to Alternaria [37], in contrast, Alt a 13 has been suggested to be another major allergen because of eliciting skin reactions in 14 of 17 patients [33]’

Point 12: Lines 117-118: It is given: “and the IgE-binding epitopes of have also been determined” – not clear epitopes of what? have also been determined, needs revision

Response 12: Thank you, the sentence has been removed.

Point 13: Line 134 says: “Unlike the above mentioned”, but it looks like there should be written, “Like the above mentioned”, check, please.

Response 13: Thank you, the sentence has been removed.

Point 14: Line 149 – word “stablish” is an obsolete variant of “establish”, better to use a newer variant.

Response 14: Thank you, the change has been done.

Point 15: Figure 1 – legend, sentence: “(e) Fragment of the molecular surface of the tetrameric 161 structure in which is shown (it is shown?) as the epitopes (epitope?), in yellow, are located buried in the structure blocking the interaction with the 162 IgE antibodies”, grammar needs revision, in general it is difficult to understand this expression

Response 15: Thank you, the sentence has been removed.

Point 16: Chapter 5, better to change its title as follows: “Immunological mechanism of Alternaria action”

Response 16: Thank you, the change has been done.

Point 17: Lines 169 – 171: grammar should be checked. It is better to say: “Within this context, it the current information about the innate and adaptive immune responses associated with Alternaria action will be summarized further with special attention to the role of Alt a 1 in allergy triggering.

Response 17: Thank you, changes have been done as suggested.

Point 18: Line 177, instead of “foreign” it is better to use “alien or external”,

Response 18: Thank you, change has been performed.

Point 19: Lines 182-183 – is given: “Similar to the mechanisms before described for aeroallergens (i.e. house dust mite) [55], Alternaria can induce the activation of epithelium…”, grammar needs revision here, no suggestion as a couple of variants are possible there, anyway, sentence is grammatically incorrect.

Response 19: Thank you, the sentence has been changed.

Point 20: Figure 2, legend: “Spores and 190 mycelium fragments containing allergenic compounds (such as Alt a 1 or proteases) can reach 191 upper airways and induce pro-inflammatory responses by means of activation (triggering? and so on) TLRs (TLR2, TLR4), PAR 2 or 192 SLC22A17 receptors present in epithelial cells

Response 20: This sentence has been changed.

Point 21: Lines 200-202, given: “heat inactivated Alternaria extracts showing that are not necessary protease activities to induce asthma pathogenesis”, better: “heat inactivated Alternaria extracts showing that protease activities are not necessary to induce asthma pathogenesis”. In case, it was mentioned something else, grammar needs more revision.

Response 21: The sentence has been changed as suggested.

Point 22: Line 203: “In the past years, more effort has been made to understand the role of different…” Probably, biological role?

Response 22: This sentence has been changed.

Point 23: Lines 206-207, it is given: “to find an explanation of the high prevalence of Alt a 1.” Prevalence where? Not clear.

Response 23: The sentence has been removed.

Point 24: Line 210 says: “In contrast, this interaction does not result in an increase in…” In the given context it is better to say: “On the other hand, this interaction does not result in an increase in”.

Response 24: The sentence has been changed as suggested.

Point 25: Line 229-300 – it is better to use “applied” instead of “exposed”.

Response 25: Thank you, the sentence has been changed.

Point 26: See more in the attached file of the article text.

Grammar of entire subchapter, within lines 254-265, should be revised.

It is recommended to add more information about the evidences that Alternaria is a primary sensitizator. You mention it in the text, but evidences are not clear.

Lines 286-287: Grammar needs revision, it is not clear, which both sources are mentioned, also, for epidemiology of thunderstorm asthma? Or some other word can be applied here, but "epidemics" seems irrelevant.

Lines 290-300 - Grammar needs revision here.

Response 26: The manuscript has been revised by the English editing service provided by the publisher.

Point 27: Fig. 3 should be given after line 324

Response 27: The change has been included.

Point 28: Lines 341-358 - What is really interesting and even funny that the only world producer of AIT treatment for Alternaria is Inmunotek company, based in Spain.

Its AIT medication, including Oraltek drug developed to treat Alternaria alternata sensitivity is successfully sold in many European countries. As far as I know, they completed their clinical and other trials and are allowed to produce their medicine.

So, with no relation to the name of company, I am sure, you should change the information given in this paragraph. Up to now this information about non-availability of AIT for Alternaria sensitization is obsolete.

Response 28: Thank you, This section has been completely updated and more information has been added.

Point 29: Lines 351-354 - Looks like this sentence fits better to the Introduction.

Response 29: The change has been included.

Round 2

Reviewer 2 Report

It is much better now, but a minor revision is still required. It should be done as follows:

Lines 97-100 write: While Alt a 1 is the only well-defined major allergen that has been recognized by skin tests in more than 90% of patients sensitized to Alternaria [35], in contrast, Alt a 13 has 98 been suggested to be another major allergen because of eliciting skin reactions in 14 of 17 99 patients [31]

However, skin test recognizes nothing; this term is applied to immunological molecular tests. So, I would recommend to change the sentence as follows or similarly: “While more than 90% of patients sensitized to Alternaria had positive skin tests to Alt a 1, it was considered the only well-defined major allergen of this source. However, Alt a 13 has been suggested to be another major allergen because of eliciting skin reactions in 14 of 17 patients”. 

Line 294 It is not very clear, but if the title is “Immunotherapy for Alternaria Asthma”, it is not very accurate too. On the one hand, mold asthma is mostly allergenic, but it is not necessary. On the other hand, AIT is generally prescribed to treat allergy, not asthma. In case, asthma is allergy-related, caused by IgE-type of allergenic sensitization, AIT would be effective as well. However, there are many classifications of asthma, not necessary caused by IgE-response. Nevertheless, the primary reason to prescribe AIT is an allergy. So, the title could be “Immunotherapy for Alternaria Sensitization”, for example.

Line 336 – you wrote: “This problem also applies to other allergens”. However, it is not necessary true as AIT applied for more than 100 years. Therefore, for many other, “old” allergens, evidences of its efficacy and safety are known and are numerous. Therefore, I would suggest removing this sentence.

Author Response

Dear Reviewer:

We are very grateful for the careful reading of our manuscript, which has allowed us to revise and clarify several parts of the text in an attempt to improve the quality of the paper. The specific points are addressed below. We have tracked changes in the manuscript marked version.

Reviewers' Comments:

REVIEWER 2:

It is much better now, but a minor revision is still required. It should be done as follows:

Lines 97-100 write: While Alt a 1 is the only well-defined major allergen that has been recognized by skin tests in more than 90% of patients sensitized to Alternaria [35], in contrast, Alt a 13 has 98 been suggested to be another major allergen because of eliciting skin reactions in 14 of 17 99 patients [31]

However, skin test recognizes nothing; this term is applied to immunological molecular tests. So, I would recommend to change the sentence as follows or similarly: “While more than 90% of patients sensitized to Alternaria had positive skin tests to Alt a 1, it was considered the only well-defined major allergen of this source. However, Alt a 13 has been suggested to be another major allergen because of eliciting skin reactions in 14 of 17 patients”. 

We agree with the comment of the reviewer. The sentence has been changed in the new revision of the manuscript in accordance with the comment

Line 294 It is not very clear, but if the title is “Immunotherapy for Alternaria Asthma”, it is not very accurate too. On the one hand, mold asthma is mostly allergenic, but it is not necessary. On the other hand, AIT is generally prescribed to treat allergy, not asthma. In case, asthma is allergy-related, caused by IgE-type of allergenic sensitization, AIT would be effective as well. However, there are many classifications of asthma, not necessary caused by IgE-response. Nevertheless, the primary reason to prescribe AIT is an allergy. So, the title could be “Immunotherapy for Alternaria Sensitization”, for example.

The title has been modified in accordance with the reviewer's indications. In this way the content of the section is more clearly reflected and ambiguities generated by the word asthma have been avoided.

Line 336 – you wrote: “This problem also applies to other allergens”. However, it is not necessary true as AIT applied for more than 100 years. Therefore, for many other, “old” allergens, evidences of its efficacy and safety are known and are numerous. Therefore, I would suggest removing this sentence.

This sentence has been removed